# Improved Composite Hydrogel for Bioengineered Tracheal Graft Demonstrates Effective Early Angiogenesis

**DOI:** 10.3390/jcm13175148

**Published:** 2024-08-30

**Authors:** Russell Seth Martins, Joanna Weber, Lauren Drake, M. Jawad Latif, Kostantinos Poulikidis, Syed Shahzad Razi, Jeffrey Luo, Faiz Y. Bhora

**Affiliations:** 1Division of Thoracic Surgery, Department of Surgery, Hackensack Meridian School of Medicine, Hackensack Meridian Health (HMH) Network, Edison, NJ 08820, USA; russellseth.martins@hmhn.org (R.S.M.); joanna.weber@gmail.com (J.W.); mohammed.latif@hmhn.org (M.J.L.); kostantinos.poulikidis@hmhn.org (K.P.); syed.razi@hmhn.org (S.S.R.); jeffrey.luo@hmhn.org (J.L.); 2Department of Surgery, Nuvance Health, Danbury, CT 06810, USA; lauren.drake@nuvancehealth.org

**Keywords:** tissue engineering, bioengineering, airways, collagen, agarose, hydrogels

## Abstract

**Background/Objectives**: Collagen–agarose hydrogel blends currently used in tracheal graft bioengineering contain relatively high concentrations of collagen to withstand mechanical stresses associated with native trachea function (e.g., breathing). Unfortunately, the high collagen content restricts effective cell infiltration into the hydrogel. In this study, we created an improved hydrogel blend with lower concentrations of collagen (<5 mg/mL) and characterized its capacity for fibroblast invasion and angiogenesis. **Methods**: Four collagen–agarose hydrogel blends were created: 1 mg/mL type 1 collagen (T1C) and 0.25% agarose, 1 mg/mL T1C and 0.125% agarose, 2 mg/mL T1C and 0.25% agarose, and 2 mg/mL T1C and 0.125% agarose. The hydrogel surface was seeded with fibroblasts, while both endothelial cells and fibroblasts (3:1 ratio) were mixed within the hydrogel matrix. We assessed early angiogenesis by observing fibroblast migration and endothelial cell morphology (elongation and branching) at 7 days. In addition, we performed immunostaining for alpha-smooth muscle actin (aSMA) and explored the gene expression of various angiogenic markers (including vascular endothelial growth factor; VEGF). **Results**: Gels with lower agarose concentrations (0.125%) with 1 or 2 mg/mL T1C were more effective in allowing early attachment and migration of surface-applied fibroblasts compared to gels with higher (0.25%) agarose concentrations. The low-agarose gels also allowed cells to quickly adopt a spread morphology and self-assemble into elongated structures indicative of early angiogenesis, while demonstrating positive immunostaining for aSMA and increased gene expression of VEGF by day 7. **Conclusions**: Hydrogel blends with collagen and low agarose concentrations may be effective in allowing early cellular infiltration and angiogenesis, making such gels a suitable cell substrate for use in the development of composite bioengineered tracheal grafts. The collagen–agarose hydrogel blend is meant to be cast around a three-dimensional (3D) printed polycaprolactone support structure and wrapped in porcine small intestine submucosa ECM to create an off-the-shelf bioengineered tracheal implant.

## 1. Introduction

There are no widely accepted materials for long-segment tracheal reconstruction, requiring the use of additional grafting materials [1]. Surgeons have therefore turned to natural and artificial grafts and autologous composites, but with limited success [2,3,4]. Tissue-engineered grafts hold promise by offering a solution for the replacement of long segments of the trachea. They can be customized for each patient and eliminate organ donor-based shortages and limitations [5]. However, our group and others have shown that pre-clinical studies demonstrate problems with graft integration, suboptimal neovascularization, and issues with mechanical strength [6,7]. The trachea is under constant mechanical duress due to pressure changes involved in regular breathing, and tracheal grafts must be able to withstand such stress [8]. Moreover, appropriate angiogenesis is key to the successful integration and long-term functioning of tissue-engineered soft tissue grafts [9], including trachea grafts [6], as cells can only remain viable at a maximum distance of 1–2 mm from a blood vessel [10]. In addition to providing nutrients to cells involved in the integration of the graft within host tissues, angiogenesis also facilitates the arrival of circulating immune and inflammatory cells that drive the tissue remodeling process during graft integration [11]. Moreover, angiogenesis is essential for mesenchymal stem cells from the bone marrow [12] to travel to the trachea to form the cartilaginous components of tracheal tissues during the graft integration process.

Normally, angiogenesis occurs in response to hypoxia [13]. Hypoxia initiates the production of nitric oxide, which in turn upregulates pro-angiogenic factors such vascular endothelial growth factor, angiopoietin 1 and 2, and nitric oxide synthase [13]. These factors work to regionally destabilize the extracellular matrix (ECM) and allow endothelial cells to proliferate and migrate towards the hypoxic area [14]. Fibroblasts are also highly motile and migrate through the ECM in response to hypoxia [15]. These fibroblasts support neovascularization and synthesize new ECM by producing matrix proteins, growth factors, proteases, and other biochemical modulators. Fibroblasts further enhance neovascularization by initiating vasodilatory expansion and improving the structural integrity of the new vasculature [15]. Co-cultures of endothelial cells and fibroblasts are commonly used in in vitro angiogenesis assays and provide an established method of testing pro-angiogenic capabilities [16].

In tissue engineering, the cell scaffold mimics the native ECM, providing a substrate for cell attachment, proliferation, and differentiation to generate new tissues [17]. Scaffolds, which can be synthetic or natural, are generally biodegradable to allow for the deposition of new and more specialized ECM to take over from the temporary scaffold structure. Hydrogels are an ideal cell scaffold for engineered soft tissues owing to their biocompatibility and mechanical and physicochemical similarity to native ECM [18]. Furthermore, hydrogels afford dimensional customizability and can be shaped by casting in molds, cutting, or 3D printing [19,20]. Previously, our group has experimented with porcine-derived small intestinal submucosa (SIS) ECM and bovine dermal ECM as the cell scaffolds of choice, with these materials being wrapped around a polycaprolactone (PCL) support structure [5,6,21,22,23]. The PCL support structure consists of solid rings (mimicking tracheal cartilage rings) with perforated, mesh-like PCL sheets connecting the solid rings. However, this approach led to dead space between the ECM cell scaffold and the PCL material, which resulted in problems with cell invasion, angiogenesis, and graft integration [5,6,21,22,23]. For these reasons, we believe that the incorporation of additional hydrogel scaffolds will augment the integration of a bioengineered composite graft with the native tracheal tissue by reducing volumetric deficit and providing an additional biocompatible substrate for cellular invasion.

For the purposes of human tracheal bioengineering, the ideal hydrogel would be composed of naturally-derived polymers with sufficient mechanical stiffness. Type 1 collagen and agarose are biopolymers that fulfill these prerequisites and are commonly used to create biocompatible and biodegradable cell scaffolds in tissue engineering. Type 1 collagen is the most abundant type of collagen in mammals and provides structure, support, and strength to connective tissues [24]. Agarose is a natural, high-molecular-weight polysaccharide with adjustable mechanical and water adsorption properties that has a wide range of applications in bioengineering and regenerative medicine [25]. Collagen–agarose blended hydrogels are able to form a co-gel with an interconnected and interpenetrating network with significantly greater storage and elastic shear modulus compared to gels of agarose and collagen separately [26]. Preliminary studies of collagen–agarose hydrogels using 0.5% agarose with 0.2% or 0.5% collagen have shown that they possess pro-angiogenic properties [27,28]. Those with 0.2% agarose and 0.2% or 0.5% collagen were found to have poor angiogenesis [27,28]. However, these studies have shown limited angiogenesis characterization, with cell morphology being the predominant mode of evaluation. Moreover, the stiffness of a 0.5% agarose and 0.5% collagen hydrogel blend is approximately 1.5 kPa [27,28], while the stiffness of mammalian tracheal soft tissue in large animal and human cadaveric studies has been shown to range from 5 to 15 kPa [29,30]. In addition, higher hydrogel concentrations lead to a decrease in porosity [31], which can negatively impact processes of cellular migration and angiogenesis [32]. Impeding these two processes within the studied hydrogel will ultimately lead to early tracheal graft failure and incomplete integration with host tissues [32]. In the design of our bioengineered composite tracheal graft, the mechanical support and structure is provided by the PCL support structure. The purpose of incorporating a hydrogel in this composite design is to eliminate the dead space between the ECM and the PCL using a material that is maximally conducive to cellular migration and angiogenesis. Higher concentrations of collagen and agarose are likely to increase the viscosity of the hydrogel blend, limiting its ability to flow and completely occupy the volumetric deficits (including permeating through the perforated inter-ring PCL sheets) in our composite bioengineered tracheal graft. Thus, we developed an improved collagen–agarose hydrogel with lower concentrations of collagen and agarose than previously reported, with the aim to decrease porosity and increase cytocompatibility. The collagen–agarose hydrogel blend is meant to be cast around a three-dimensional (3D) printed polycaprolactone support structure and wrapped in porcine small intestine submucosa ECM to create an off-the-shelf bioengineered tracheal implant (Appendix A). In this study, we characterize the capacity for fibroblast invasion and angiogenesis in this improved lower-concentration (compared to 5 mg/mL in the aforementioned literature) gel blend.

## 2. Materials and Methods

The protocol received ethical approval from the institutional review board (Reference # 2019-19). No human or animal subjects were involved in this study.

### 2.1. Cell Culture

Human umbilical vein endothelial cells and dermal fibroblasts were purchased from the American Type Culture Collection (ATCC), Manassas, Virginia, United States of America. ATCC-pooled HUVECs are derived from 10 individual donors, thus lending increased generalizability of results as compared to a single donor. Cells were expanded in two dimensions (2D) in vascular growth and low serum fibroblast growth kits (ATCC) with media exchange every 2–3 days. The presence of 2% fetal bovine serum in the low-serum fibroblast growth kits supports more prolific growth compared to the serum-free medium. Cells within 7 passages were used. No morphological changes were observed.

### 2.2. Hydrogel Construct Creation

Stock molten agarose solutions of 1% weight/volume (*w*/*v*) and 0.5% *w*/*v* were created by heating sterilized low-melt agarose powder (Sigma Aldrich, St. Louis, MI, USA) to 60 °C on a stir plate. A stock neutralized collagen solution (Sigma Aldrich, St. Louis, MI, USA) of 10 mg/mL was created according to the manufacturer’s instructions.

The gel blends created were the following: 1 mg/mL T1 collagen with 0.25% agarose, 1 mg/mL T1 collagen with 0.125% agarose, 2 mg/mL T1 collagen with 0.25% agarose, and 2 mg/mL T1 collagen with 0.125% agarose. An amount of 200 µL of each gel blend was added to the wells of a 48-well plate, with five replicates per gel blend being used. The gelation was conducted first at 4 °C for 10 min and then at 37 °C for 30 min. Fibroblasts were trypsinized, counted, and suspended at a concentration of 10^6^ cells/mL in the fibroblast growth medium. An amount of 200 µL of cell suspension (corresponding to 2 × 10^5^ cells) was applied to the surface of the hydrogel constructs. Medium was exchanged on day 4 to allow sufficient time for the cells to remain in contact with the gel surface undisturbed.

Endothelial cells were trypsinized, counted, and mixed in a 3:1 ratio of fibroblasts to endothelial cells. The cell mix was pelleted and mixed with the prepared hydrogel blends, resulting in a final overall cell concentration of 1–3 × 10^6^ cells/mL. Then, 300 µL of each cell-gel mixture was pipetted into the wells of a 24-well plate. After gelation (10 min at 4 °C and 30 min at 37 °C), 500 µL of complete vascular growth medium was added to the gels, and cultures were maintained at 37 °C for 7 days with media exchange every 2–3 days.

Cell morphology images were captured using a Vectra automated imaging system (Akoya Biosciences, Marlborough, MA, USA). Analysis was performed by a blinded observer who made qualitative observations of the images. Key parameters included comparative assessment of cell length (i.e., contiguous areas that are brighter/darker than the surrounding background) and elongation (i.e., deviation from spherical cell shape).

### 2.3. Assessment of Fibroblast Migration and Angiogenesis

After 7 days of culture, the medium was removed, and gels were preserved in either formalin or RNAlater^®^ (Thermo Fisher Scientific Inc., Waltham, MA, USA) or snap-frozen in liquid nitrogen and stored at −80 °C. RNAlater^®^ is an RNA stabilization and storage agent that minimizes the need to immediately process or freeze tissue samples without jeopardizing the quality or quantity of RNA. The formalin-fixed constructs were cut in half to expose vertical cross sections and processed for paraffin embedding. Then, 5 µm sections were cut, and the average depth of fibroblast invasion was measured on phase-contrast imaging on days 1, 4, and 7 after seeding.

We initially prepared three slides per gel blend. We selected one cross-section slide per gel blend for visualization and analysis based on quality of tissue section and number of cells present. Image sections of fibroblast migration into the hydrogel blends were imported into ImageJ 1.52a (U.S. National Institutes of Health, Bethesda, MD, USA) and converted to a 32-bit image. Cells were identified via intensity thresholding against the light hydrogel background. Afterward, the densely populated cell seeding surface was segmented from migrated cells by identifying large contiguous cell areas (i.e., greater than 15 µm^2^) indicating the contours of the collagen–agarose hydrogels. The distances between migrated cells and the nearest seeding surface was calculated using an ImageJ macro [33]. Endothelial cell differentiation was responsible for angiogenesis, and the degree of endothelial cell elongation is correlated with angiogenic.

Select slides (i.e., sections with minimum defects) were also immunostained for alpha-smooth muscle actin. Sections were incubated overnight at 4 °C in the primary antibody (anti-aSMA; 1:100 dilution), followed by 2 h room temperature application of the 1:200 FITC (fluorescein isothiocyanate)-conjugated secondary antibody. Slides were visualized using a Vectra automated imaging system (Akoya Biosciences: Marlborough, MA, USA).

RNA was purified using the RNeasy Mini Spin Columns (Qiagen, Inc., Hilden, Germany) according to the manufacturer’s instructions and eluted in 10 μL of nuclease-free water. RNeasy Mini Spin Columns separate RNA from other biological material using a silica membrane. RNA quantity and quality was measured using the NanoDrop^®^ (Thermo Fisher Scientific Inc., Waltham, MA, USA) instrument. Complementary DNA (cDNA) was synthesized from RNA using the SuperScript™ VILO™ Master Mix (Thermo Fisher Scientific Inc., Waltham, MA, USA) with 2.5 µg of RNA, according to the manufacturer’s instructions. qPCR was run on a QuantStudio™ 7 instrument (Thermo Fisher Scientific Inc., Waltham, MA, USA) with standard SYBR™ Green (Thermo Fisher Scientific Inc., Waltham, Massachusetts, United States of America) chemistry using the primers to probe for genes shown in Table 1. Beta-2 microglobulin was used as the housekeeping gene and served to standardize the results of the RNA analysis. Biological and technical replication was conducted by repeat experimentation.

### 2.4. Statistical Analysis

The average cell migration distances in the different hydrogels at days 1, 4, and 7 were compared for statistical significance using the Mann–Whitney U test. The average fold change (day 7 compared to day 1) in gene expression for the genes shown in Table 1 was compared between the four hydrogel blends using the Kruskal–Wallis test, with post hoc analysis being performed using Dunn’s test. All statistical analysis was performed using OriginLab (OriginLab Corporation, Northampton, MA, USA) and IBM SPSS Statistics for Windows, version 22 (IBM Corp., Armonk, NY, USA). A *p*-value < 0.05 was considered significant for all analyses.

## 3. Results

### 3.1. Fibroblast Migration

Hydrogel constructs with agarose concentrations of 0.125% most effectively allowed surface-applied fibroblasts to attach and migrate into the gel constructs as early as day 1 post-seeding. In contrast, hydrogel constructs with agarose concentrations of 0.25% demonstrated inferior fibroblast attachment and no migration at all timepoints, with seeded cells remaining clumped and not infiltrating the gel (average migration distance~0 µm).

On comparing cell migration distances between the two 0.125% agarose blends, the 2 mg/mL collagen blend showed significantly higher cell migration distances than the 1 mg/mL blend (2 mg/mL collagen: 52.5 ± 46.4 µm vs. 1 mg/mL collagen: 29.7 ± 34.0 µm; *p* = 0.001). However, this difference did not persist at days 4 and 7, with both the 2 mg/mL and 1 mg/mL collagen blends showing comparable migration distances. These data are shown in Figure 1 and Figure 2 and Table 2. In addition, the demonstration of fibroblast migration within the hydrogel matrix indicates the following regarding the collagen–agarose blend: (1) it is biocompatible; (2) it displays cell attachment motifs to allow cell motility; and (3) it remains sufficiently porous to permit nutrient diffusion and cell infiltration.

### 3.2. Angiogenesis

Hydrogel constructs with agarose concentrations of 0.125% most effectively allowed endothelial cells to quickly adopt a spread morphology and self-assemble into elongated structures (i.e., deviating from spherical cell shapes) indicative of early vasculogenesis. More specifically, hydrogels composed of 0.125% agarose and 1–2 mg/mL T1 collagen produced the greatest (“extensive”) cell elongation based on blinded comparative analysis.

Amongst the constructs with 0.25% agarose, the hydrogel with 1 mg/mL T1 collagen indicated greater potential for early vasculogenesis (i.e., minimal cell extension) than the hydrogel with 2 mg/mL T1 collagen (i.e., no cell extensions observed) (Figure 3). Of note, both 0.25% hydrogel formulations produced significantly reduced and inconsistent (“minimal”) cell elongation compared to the 0.125% hydrogel formulations, as assessed during blinded comparative analysis.

In addition, the demonstration of early vasculogenesis in select collagen–agarose concentrations further supports key hydrogel biological and mechanical characteristics, including biocompatibility (the majority of endothelial cells remain viable enough to undergo vasculogenesis), cell attachment motifs (it permits cells to form focal adhesions during the elongation and vasculogenesis process), and porosity (it ensures sufficient nutrient/waste exchange occurs during the 7-day culture period). Figure 4 and Table 3 show the day 7 change in the expression of various genes in the different hydrogel constructs compared to day 0 controls. Constructs with 0.125% agarose concentration showed greater fold increases in the expression of CDH5, ACTA2, and VEGF by day 7 compared to those with a 0.25% agarose concentration (*p* < 0.001). In fact, the 2 mg/mL T1 collagen and 0.25% agarose concentration gel showed decreased expression of CDH5 (fold decrease: 1.52 times) and ACTA2 (fold decrease: 1.32 times) by day 7, while levels of FTL and VEGF were undetectable. Both constructs with 0.125% agarose and the construct with 1 mg/mL T1 collagen and 0.25% agarose concentration showed decreased expression of TIE1 and FTL by day 7.

Post hoc analysis comparing the 0.125% agarose concentration blends revealed that compared to the 2 mg/mL collagen blend, the 1 mg/mL T1 collagen blend had a significantly greater fold increase in CDH5 (fold increase: 12.13 times vs. 10.27 times; *p* < 0.001), ACTA2 (47.5 times vs. 18.64 times; *p* < 0.001), and VEGF (12.13 times vs. 2.48 times; *p* < 0.001). The fold decrease in FTL was greater in the 2 mg/mL collagen blend and the 1 mg/mL blend (fold decrease: 2.17 times vs. 43.47 times; *p* < 0.001). There was no significant difference in the fold decrease in TIE1 between the 1 mg/mL collagen blend and the 2 mg/mL blend (fold decrease: 3.70 times vs. 3.45 times; *p* = 0.08).

The hydrogel blends with agarose concentrations of 0.125% demonstrated positive immunostaining for aSMA (Appendix A).

## 4. Discussion

There has been limited characterization of the angiogenic potential of collagen–agarose blended hydrogels, restricting their use in tissue engineering. We created an improved collagen–agarose hydrogel blend with lower concentrations of both biopolymers than previously studied [27,28] and tested its angiogenic properties. Our results revealed that for these relatively collagen-rich hydrogel blends, a lower agarose content (0.125%) is most effective in allowing cellular attachment, invasion, and angiogenesis. These improved hydrogel blends retained their structural integrity and did not undergo any noticeable degradation or contracture during the 7-day study period, confirming their suitability to act as a filler material between the porcine ECM and the PCL scaffold in our composite tracheal graft.

Early, appropriate angiogenesis is a key component in the successful implantation of a bioengineered tracheal graft as it allows for the delivery of adequate nutrition, the arrival of inflammatory cells with a role in remodeling, and bone marrow-derived mesenchymal stem cell seeding [6,9,10,11,12]. The angiogenic properties of a hydrogel depend on several factors, such as its composition, stiffness, and microstructure [34]. Since these characteristics are often interrelated, the development of blended hydrogels must balance the intricate interplay between specific requirements for mechanical strength and cytocompatibility [34]. Previous studies of collagen–agarose hydrogels using 0.5% (5 mg/mL) agarose with 0.2% (2 mg/mL) or 0.5% (5 mg/mL) collagen showed that they possess pro-angiogenic properties [27,28], while blends with 0.2% agarose were found to have poor angiogenesis [27,28]. Overly dense hydrogel blends are less porous and less conducive to cell migration and biomolecular movement through the hydrogel matrix [31,35], thus limiting cellular behavior and communication and critically reducing vasculogenic potential. Relatively collagen-rich hydrogels may also aid in cellular movement by providing a network for guidance and signaling. Conversely, insufficiently concentrated hydrogels are prone to mechanical failure, necessitating studies to optimize the compromise between structural integrity and nutrient exchange via vascularization. Interestingly, the best potential for angiogenesis and cell invasion was seen in blends with a lower concentration of agarose (0.125%). These results seemingly contradict previous studies, where satisfactory angiogenic potential was demonstrated in hydrogels with 0.5% agarose concentrations [27]. However, it is likely that the increased concentrations of agarose used increase the density of the hydrogel (and thus lower the porosity), and thus lower concentrations of agarose are needed to achieve a balance. Lastly, less porous and thus stiffer hydrogels would also not suit their intended purpose in our composite bioengineered tracheal graft (i.e., to act as a filler material to occupy dead space and promote more continuous cell migration). Thus, the low-agarose hydrogel blends seem to be the best fit for tracheal tissue engineering as they strike the optimum balance of physical and chemical properties.

Optimizing cell infiltration in bioengineered grafts allows cells to enter scaffold materials to remodel hydrogel matrices, produce extracellular matrix, support vascularization, and initiate the critical process of graft integration [36]. A variety of characteristics of a collagen–agarose hydrogel blend impacts its conduciveness to fibroblast migration, including porosity and pore size, mechanical stiffness, adhesion ligands and moieties (e.g., arginine-glycine-aspartic acid or RGD on collagen), degradation rate, water content and swelling properties, and biocompatibility (i.e., lack of cellular toxicity) [37]. Our results demonstrated that the low-agarose-concentration gel blends (0.125% agarose) were most conducive to cell infiltration, with fibroblast invasion being seen as early as day 1. On the contrary, the 0.25% agarose gel blends did not show any meaningful fibroblast invasion [37]. When comparing the migration distances between the two 0.125% agarose gel blends (i.e., 1 mg/mL and 2 mg/mL collagen), no significant differences were found in the extent of fibroblast invasion at days 4 and 7. Interestingly, the 2 mg/mL collagen blend showed significantly greater fibroblast migration on day 1 compared to the 1 mg/mL blend. This suggests that the higher collagen content promotes an earlier onset of fibroblast invasion, possibly due to the denser concentration of cell adhesion ligands or moieties, such as RGD. Ultimately, however, both 0.125% agarose gels were able to achieve comparable fibroblast invasion and migration by day 7.

Our results also showed increased expression of VEGF, CDH5, and ACTA2 in both hydrogel constructs with 0.125% agarose and decreased expression of TIE1 and FTL. VEGF is a major growth factor for endothelial cell and plays a prominent role in supporting angiogenesis by stimulating the proliferation, migration, and survival of endothelial cells [38,39,40]. Cadherin, coded by CDH5, serves as an endothelial junction protein, where it regulates cell proliferation and modulates VEGF receptor functions during periods of angiogenesis [41,42]. This modulation of the VEGF receptor is especially important in airway bioengineering, as excessive VEGF activity stimulates endothelial overproliferation, uncontrolled collagenation, and eventual fibrosis and airway stenosis [43]. Moreover, the increased expression of ACTA2 was corroborated by the visualization of aSMA on immunostaining. Although the 47.5-fold increase in ACTA2 is considerably large, the magnitude of change is within the ranges that have been documented in the previous literature. Fibroblast expression of aSMA is implicated in the remodeling of the extracellular matrix [44], which is a critical step in angiogenesis as it allows cellular migration [14].

Our results showed that the hydrogel blends with 0.125% agarose concentration showed significantly superior expression of CDH5, ACTA2, and VEGF by day 7 compared to those with a 0.25% agarose concentration. In contrast, the 2 mg/mL T1 collagen and 0.25% agarose concentration gel showed decreased expression of CDH5 and ACTA2 by day 7, while levels of FTL and VEGF were undetectable. This pattern of gene expression corroborates our qualitative microscopic analysis and likely underscores how reduced porosity in high-density gels (2 mg/mL T1 collagen and 0.25% agarose) suppresses endothelial cell spreading, elongation, and angiogenesis. Interestingly, both constructs with 0.125% agarose and the construct with 1 mg/mL T1 collagen and 0.25% agarose concentration showed decreased expression of TIE1 and FTL by day 7. The lower expression of TIE1 in hydrogels demonstrating early angiogenesis may be due to its more prominent role in the later stages of angiogenesis, particularly endothelial cell maturation and quiescence [45], that were not captured in our experiments. Lastly, while the expression of the FTL gene is increased during angiogenesis in vivo, its expression was low in hydrogels with successful early angiogenesis in our study. Given that the major role of ferritin is to antagonize endogenous inhibitors of angiogenesis, such as cleaved high-molecular-weight kininogen, the absence of such inhibitors in an in vitro system may reduce the amount of FTL gene expression necessary for successful angiogenesis [46]. Lastly, compared to the 0.125% agarose and 2 mg/mL T1 collagen blend, the 1 mg/mL T1 collagen blend had a significantly greater fold increase in CDH5 and VEGF and a smaller fold decrease in FTL. These findings suggest that lower concentrations of collagen in combination with lower combinations of agarose are most suitable for angiogenesis. This is presumably a function of the increased porosity of the hydrogel blend, which allows for better endothelial cell spreading and blood vessel morphogenesis [47]. Taken together with the comparable capacity for fibroblast cell migration in the 1 and 2 mg/mL T1 collagen and 0.125% agarose gels, it would seem that the 0.125% agarose and 1 mg/mL T1 collagen gel blend is the most suitable candidate for use in our composite tracheal graft construct.

Our study has a few limitations. First, we tested limited combinations of collagen and agarose. Second, our assessment of angiogenesis on histology and immunostaining was qualitative. However, our results lay important foundations for future work that should characterize the rheological properties of low-agarose hydrogel blends, explore their porosity, rate of degradation, and swelling under in vivo conditions, and assess their structural suitability for use in composite tracheal grafts.

## 5. Conclusions

Hydrogel blends with lower concentrations of collagen and agarose are most effective in allowing cellular attachment and migration and angiogenesis as early as the first 7 days. This may be attributed to the increased porosity of less concentrated gel blends, which facilitates cellular migration. These properties suggest that low-agarose hydrogel blends are a suitable cell substrate for use in the development of composite bioengineered tracheal grafts.

## Figures and Tables

**Figure 1 jcm-13-05148-f001:**
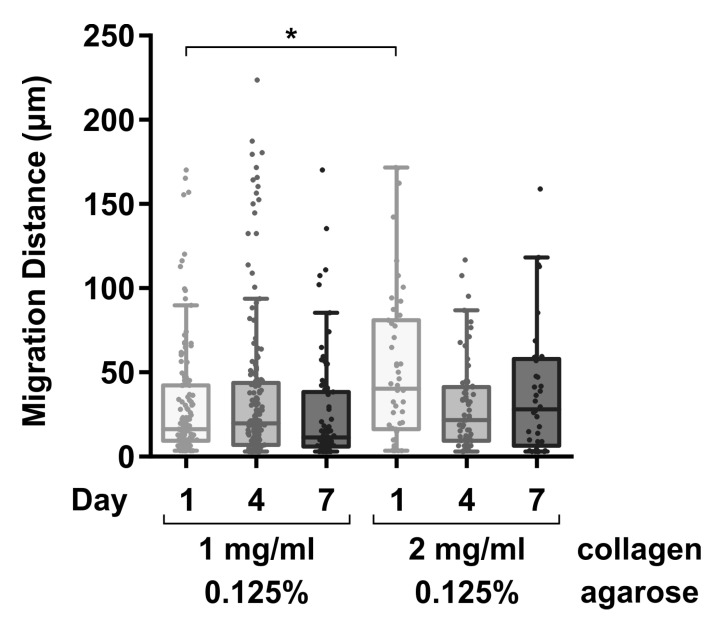
Microscopic images of fibroblast migration into hydrogel constructs. * indicates *p* < 0.05 between 1 mg/mL and 2 mg/mL collagen blends at day 1.

**Figure 2 jcm-13-05148-f002:**
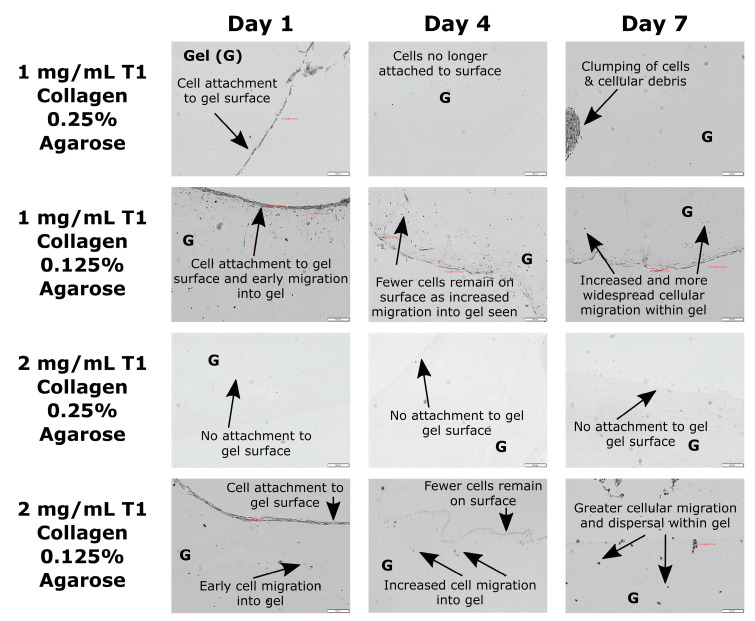
Cell migration in the collagen–agarose hydrogel blends over days 1–7.

**Figure 3 jcm-13-05148-f003:**
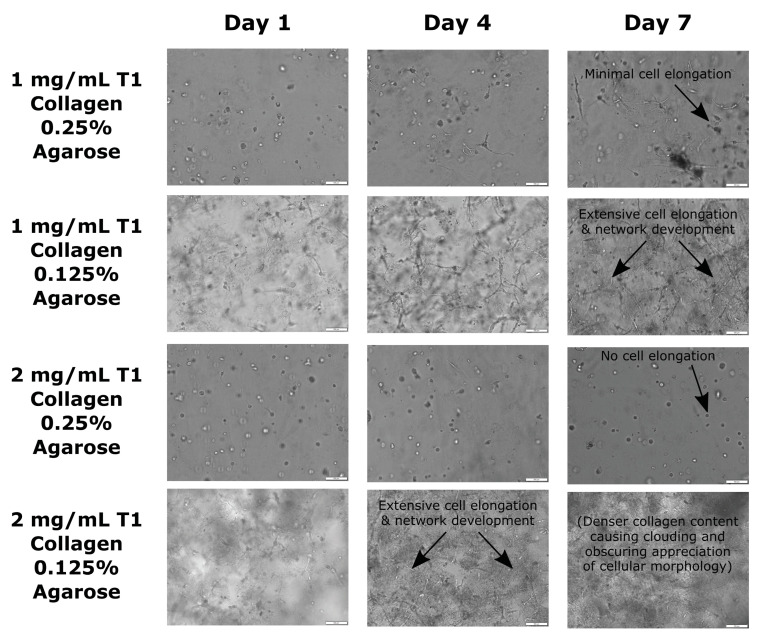
Microscopy images of cell morphology among the different hydrogel blends.

**Figure 4 jcm-13-05148-f004:**
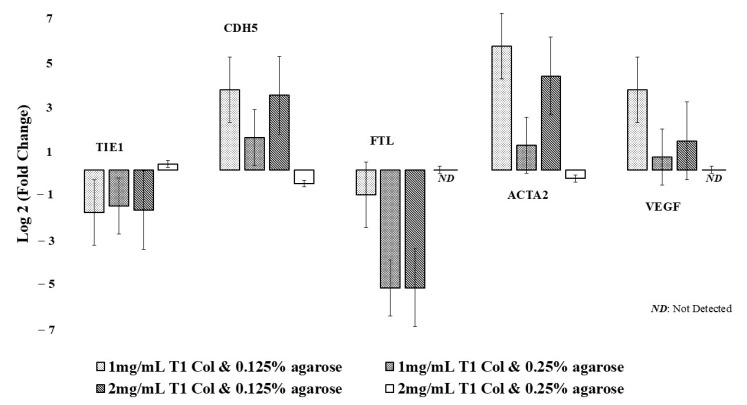
Gene expression across different hydrogel blends.

**Table 1 jcm-13-05148-t001:** Primers.

Gene	Forward	Reverse
*TIE1*	ATGGCTGCTCTTGTGGATCTGG	CGGTCACAAGTGCCACCATTCT
*CDH5*	GAAGCCTCTGATTGGCACAGTG	TTTTGTGACTCGGAAGAACTGGC
*FTL*	TACGAGCGTCTCCTGAAGATGC	GGTTCAGCTTTTTCTCCAGGGC
*ACTA2*	CTATGCCTCTGGACGCACAACT	CAGATCCAGACGCATGATGGCA
*VEGF*	TTGCCTTGCTGCTCTACCTCCA	GATGGCAGTAGCTGCGCTGATA

*ACTA2:* Actin alpha 2, smooth muscle; *CDH5:* Cadherin 5; *FTL:* Ferritin light chain; *TIE1*: Tyrosine kinase with immunoglobulin -like and EGF-like domains 1; *VEGF*: Vascular endothelial growth factor.

**Table 2 jcm-13-05148-t002:** Maximum cumulative migration distances of fibroblasts within the hydrogel constructs.

Hydrogel Composition	1 mg/mL T1 Collagen & 0.125% Agarose	2 mg/mL T1 Collagen & 0.125% Agarose	*p*-Value
Day 1 (Mean ± SD/Median [Q1–Q3])	29.7 ± 34.0 µm 16.2 [9.2–42.0] µm	52.5 ± 46.4 µm 40.4 [16.2–80.6] µm	0.001
Day 4 (Mean ± SD/Median [Q1–Q3])	35.8 ± 44.6 µm 19.5 [6.9–43.9] µm	29.4 ± 26.9 µm 21.5 [9.3–41.2] µm	0.836
Day 7 (Mean ± SD/Median [Q1–Q3])	25.3 ± 32.6 µm 11.4 [5.4–38.9] µm	38.5 ± 38.9 µm 28.1 [5.0–58.2] µm	0.073

**Table 3 jcm-13-05148-t003:** Changes in gene expression across different hydrogels.

Genes		1 mg/mL T1 Collagen & 0.25% Agarose	1 mg/mL T1 Collagen & 0.125% Agarose	2 mg/mL T1 Collagen & 0.25% Agarose	2 mg/mL T1 Collagen & 0.125% Agarose	*p*-Value
*TIE1*	Fold Change	−3.03	−3.70	+1.21	−3.45	<0.001
	Log_2_ (Fold Change)	−1.61	−1.91	+0.27	−1.80	
*CDH5*	Fold Change	+2.75	+12.13	−1.52	+10.27	<0.001
	Log_2_ (Fold Change)	+1.46	+3.60	−0.61	+3.36	
*FTL*	Fold Change	−43.47	−2.17	0% *	−43.47	<0.001
	Log_2_ (Fold Change)	−5.30	−1.12	-	−5.30	
*ACTA2*	Fold Change	+2.17	+47.50	−1.32	+18.64	<0.001
	Log_2_ (Fold Change)	+1.12	+5.57	−0.39	+4.22	
*VEGF*	Fold Change	+1.52	+12.13	0% *	+2.48	<0.001
	Log_2_ (Fold Change)	+0.60	+3.60	-	+1.31	

* Not detected at both timepoints.

## Data Availability

The raw data supporting the conclusions of this article will be made available by the authors on request.

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
