# Peer review of "Improved Composite Hydrogel for Bioengineered Tracheal Graft Demonstrates Effective Early Angiogenesis"

_jcm, 2024, doi:10.3390/jcm13175148_

Round 1

Reviewer 1 Report

Comments and Suggestions for Authors

The manuscript explores a novel hydrogel composite comprising Collagen and agarose for tracheal graft bioengineering. Below are suggestions aimed at enhancing the study's quality:

Abstract Section:

The authors adeptly highlight the challenge posed by low collagen concentration in existing collagen-agarose composites, leading to mechanical failures (line 12). However, to truly illuminate the potential of the reported composite, it would be wonderful to see data on rheological properties, mechanical characteristics, and insights into hydrogel swelling, degradation, and porosity variations.

Introduction Section:

-            I suggest the authors to explore the rate of degradation of the hydrogel composites concerning varying cell seeding densities.

Lines 89-90 raise intriguing questions as the literature reported seems to diverge from the study findings. Exploring this contradiction in the discussion with compelling data could be enriching.

-            A reminder to double-check the wording in line 109 regarding "decrease porosity."

Methods Section:

- Let's ensure that cell density is clearly stated as 10^6 in lines 131 and 132 for seamless comprehension.

-            Mentioning the number of replicates used for each condition would be effective

-            Providing clarity on the absence of results at the 14-day time point (line 140) would enhance understanding.

-            Including company and reagent details in the main text would illuminate the process for readers(for example, RNAlater).

-            Please relocate lines 165-171 from the "Statistical Analysis" subsection.

Results Section:

-            Please improve figure quality and descriptions [ example: Mention the scale bar for figure 1 in the description]

-            Figure 2: mention the number of cross-sections studied per sample and the statistical analysis method employed

-            Given the significance of porosity, including ESEM images for all composites would provide valuable insights.

-            Deepen the discussion on the rate of vascularization with different composites.

As the authors themselves mentioned, the number of Collagen-agarose combinations mentioned here is not enough to increase the confidence that increasing the collagen concentration would better the tracheal graft bioengineering. I suggest the authors include more combinations for impactful arguments.

Author Response

­­Dear Editor,

Thank you and the reviewers for making the time to review our manuscript and providing us with valuable feedback. We have made extensive revisions to our manuscript based on the comments received from the reviewers. We believe these changes have substantially improved the quality of our manuscript.

Below is a point-by-point response to all of the comments received:

*          *          *          *          *

Reviewer 1

Comment 1: The manuscript explores a novel hydrogel composite comprising Collagen and agarose for tracheal graft bioengineering. Below are suggestions aimed at enhancing the study's quality:

Authors’ response: Thank you for taking the time to review our manuscript and for providing us with constructive feedback to elevate the quality of our manuscript. All Authors’ have discussed the comments received and we have made all the necessary and possible revisions to our manuscript.

Comment 2: The Authors adeptly highlight the challenge posed by low collagen concentration in existing collagen-agarose composites, leading to mechanical failures (line 12). However, to truly illuminate the potential of the reported composite, it would be wonderful to see data on rheological properties, mechanical characteristics, and insights into hydrogel swelling, degradation, and porosity variations. I suggest the Authors’ explore the rate of degradation of the hydrogel composites concerning varying cell seeding densities. Given the significance of porosity, including ESEM images for all composites would provide valuable insights.

Authors’ response: Thank you for your comment. We agree that mechanical characterization is important and interesting to explore. However, the ultimate purpose of the hydrogel is not mechanical but rather to augment the integration of a bioengineered composite graft with the native tracheal tissue by reducing volumetric deficit and providing an additional biocompatible substrate for cellular invasion.

Secondly, although beyond the scope of the present manuscript, our current iteration of experiments is exploring the performance of the hydrogel when used with the composite trachea graft (i.e., extracellular matrix [ECM] wrapped around a polycaprolactone [PCL] support structure, with the hydrogel layered between the ECM and the PCL to reduce volumetric deficit and make the graft more conducive to cellular migration). These findings will report on the performance of the hydrogels when used for their specific, intended purpose, and will be published in a subsequent manuscript. Thank you for suggesting the exploration of hydrogel swelling and porosity – we will ensure to perform these experiments in a subsequent paper. Encouraging, at this point in our research, there was no gross degradation of the hydrogels during the present study experiment.

We added the following to our Limitations paragraph in the Discussion: “Our study has a few limitations. First, we tested limited combinations of collagen and agarose. Secondly, our assessment of angiogenesis on histology and immunostaining was qualitative. However, our results lay important foundations for future work that should characterize the rheological properties of low-agarose hydrogel blends, explore their porosity, rate of degradation, swelling under in vivo conditions, and assess their structural suitability for use in composite tracheal grafts.”

Comment 3: Lines 89-90 raise intriguing questions as the literature reported seems to diverge from the study findings. Exploring this contradiction in the discussion with compelling data could be enriching.

Authors’ response: Thank you for this important suggestion. We have expanded on our Discussion by adding the following: “Previous studies of collagen-agarose hydrogels using 0.5% (5 mg/mL) agarose with 0.2% (2 mg/mL) or 0.5% (5 mg/mL) collagen have shown that they possess pro-angiogenic properties (27, 28), while blends with 0.2% agarose are found to have poor angiogenesis (27, 28). Overly dense hydrogel blends are less porous and less conducive to the cell migration and biomolecular movement through the hydrogel matrix (31, 35), thus limiting cellular behavior and communication, and critically reducing angiogenic potential. Relatively collagen-rich hydrogels may also aid in cellular movement by providing a network for guidance and signaling. . Conversely, insufficiently concentrated hydrogels are prone to mechanical failure, necessitating studies to optimize the compromise between structural integrity and nutrient exchange via vascularization. Interestingly, the best potential for angiogenesis and cell invasion was seen in blends with a lower concentration of agarose (0.125%). These results seemingly contradict previous studies, where satisfactory angiogenic potential was demonstrated in hydrogels with 0.5% agarose concentrations (27). However, it is likely that the increased concentrations of agarose used increases the density of the hydrogel (and thus lowers the porosity), and thus lower concentrations of agarose are needed to achieve a balance. Lastly, less porous and thus stiffer hydrogels would also not suit their intended purpose in our composite bioengineered tracheal graft (i.e., to act as a filler material to occupy dead space and promote more continuous cell migration). Thus, the low-agarose hydrogel blends seem to be the best fit for tracheal tissue engineering, as they strike the optimum balance of physical and chemical properties.”

Comment 4: A reminder to double-check the wording in line 109 regarding "increase porosity."

Authors’ response: Thank you for noticing this error. We have fixed it.

Comment 5:  Let's ensure that cell density is clearly stated as 10^6 in lines 131 and 132 for seamless comprehension.

Authors’ response: Thank you for noticing this error. We have fixed it.

Comment 6: Mentioning the number of replicates used for each condition would be effective.

Authors’ response: Thank you for this suggestion. We have added the following in our Materials and Methods: “The gel blends created were the following: 1 mg/mL T1 collagen with 0.25% agarose, 1 mg/mL T1 collagen with 0.125% agarose, 2 mg/mL T1 collagen with 0.25% agarose, and 2 mg/mL T1 collagen with 0.125% agarose. 200µL of each gel blend was added to the wells of a 48-well plate, with five replicates per gel blend being used.”.

Comment 7: Providing clarity on the absence of results at the 14-day time point (line 140) would enhance understanding.

Authors’ response: Thank you for noticing our error. We only maintained our seeded hydrogels for 7 days, after which we preserved, processed, or froze them. We have fixed this in the Materials and Methods.

Comment 8: Including company and reagent details in the main text would illuminate the process for readers(for example, RNAlater).

Authors’ response: Thank you for your suggestion. We have added the following to our Materials and Methods:

“Human umbilical vein endothelial cells and dermal fibroblasts were purchased from the American Type Culture Collection (ATCC), Manassas, Virginia, United States of America. ATCC pooled HUVECs are derived from 10 individual donors, thus lending increased generalizability of results as compared to a single donor. Cells were expanded in two dimensions (2D) in vascular growth and low serum fibroblast growth kits (ATCC) with media exchange every 2-3 days. The presence of 2% fetal bovine serum in the low serum fibroblast growth kits supports more prolific growth compared to the serum-free medium. Cells within 7 passages were used. No morphological changes were observed.”

“After 7 days of culture, the medium was removed and gels were preserved in either formalin or RNAlater® (Thermo Fisher Scientific Inc., Waltham, Massachusetts, United States of America), or snap frozen in liquid nitrogen and stored at -80°C. RNAlater® is an RNA stabilization and storage agent that minimizes the need to immediately process or freeze tissue samples, without jeopardizing the quality or quantity of RNA.”.

“RNA was purified using the RNeasy Mini Spin Columns (Qiagen, Inc., Hilden, Germany) according to the manufacturer’s instructions and eluted in 10uL of nuclease-free water. RNeasy Mini Spin Columns separate RNA from other biological material using a silica membrane. RNA quantity and quality was measured using the NanoDrop® (Thermo Fisher Scientific Inc., Waltham, Massachusetts, United States of America) instrument. Complementary DNA (cDNA) was synthesized from RNA using the SuperScript™ VILO™ Master Mix (Thermo Fisher Scientific Inc., Waltham, Massachusetts, United States of America) with 2.5µg of RNA according to the manufacturer’s instructions. qPCR was run on a QuantStudio™ 7 instrument (Thermo Fisher Scientific Inc., Waltham, Massachusetts, United States of America) with standard SYBR™ Green (Thermo Fisher Scientific Inc., Waltham, Massachusetts, United States of America) chemistry using the primers to probe for genes shown in Table 1. Beta-2 microglobulin was used as the housekeeping gene and served to standardize the results of the RNA analysis.”.

Comment 9: Please relocate lines 165-171 from the "Statistical Analysis" subsection.

Authors’ response: Thank you for your suggestion. We have relocated those lines from the “Statistical Analysis” subsection to the “Assessment of Fibroblast Migration and Angiogenesis” subsection.

Comment 10: Please improve figure quality and descriptions [example: Mention the scale bar for figure 1 in the description]

Authors’ response: Thank you. We have improved the figure quality and added scale bars.

Comment 12: Figure 2: mention the number of cross-sections studied per sample and the statistical analysis method employed.

Authors’ response: Thank you for inquiring further into our methodology. We initially prepared three slides per gel blend. We selected one cross-section slide per gel blend for visualization and analysis (slide selected based on quality of tissue section and number of cells present). Cell migration distances were analyzed using the Mann-Whitney U test. We have mentioned this in our Methods: “We initially prepared three slides per gel blend. We selected one cross-section slide per gel blend for visualization and analysis, based on quality of tissue section and number of cells present.”.

Comment 12: Deepen the discussion on the rate of vascularization with different composites.

Authors’ response: Thank you for suggesting this point for deepening discussion. We have supplemented the Discussion section to address key features which affect the rate of vascularization (e.g., concentration/density) and how these may affect other pertinent hydrogel behaviors (e.g., structural integrity)

The following text has been included in the manuscript

“Overly dense hydrogel blends are less porous and less conducive to the cell migration and biomolecular movement through the hydrogel matrix (31, 35), thus limiting cellular behavior and communication, and critically reducing vasculogenic potential. Conversely, insufficiently concentrated hydrogels are prone to mechanical failure, necessitating studies to optimize the compromise between structural integrity and nutrient exchange via vascularization.”

Comment 15: As the Authors’ themselves mentioned, the number of collagen-agarose combinations mentioned here is not enough to increase the confidence that increasing the collagen concentration would better the tracheal graft bioengineering. I suggest the Authors’ include more combinations for impactful arguments.

Authors’ response: Thank you for your comment. Collagen has limited solubility at neutral pH. While we could increase the solubility, it would involve either acidifying the solution or altering its temperate, both of which would have been detrimental to cell viability.

Reviewer 2

Comment 1: The Article is devoted to the study of novel collagen-agarose-based hydrogel blends which will help in the development of bioengineered tracheal grafts. The study is interesting, important, and great efforts were put into its realization. Authors’ studied the newly created materials for fibroblast invasion and angiogenesis for 7 days. Interesting that the co-cultures of human umbilical vein endothelial cells and dermal fibroblasts were used during the research. The limitations of the study are characterized, which is a benefit. The following comments do not diminish the value of the Article.

Authors’ response: Thank you for taking the time to review our manuscript, your appreciation of our manuscript, and for providing us with constructive feedback to elevate the quality of our manuscript. All Authors’ have discussed the comments received and we have made all the necessary and possible revisions to our manuscript.

Comment 2: Line 3 Probably it would be better to specify in the Article the criteria that might be used to assess the effectiveness of angiogenesis. As this term is used in the title of the Article.

Authors’ response: Thank you for your suggestion. We have added the following to the Abstract: “We assessed early angiogenesis by observing fibroblast migration and endothelial cell morphology (elongation and branching) at 7 days. In addition, we performed immunostaining for alpha smooth muscle actin (aSMA) and explored the gene expression of various angiogenic markers (including vascular endothelial growth factor; VEGF).”.

Comment 6: Would you please specify why the current concentrations of collagen (1 mg/mL and 2 mg/mL) and agarose (0.125% and 0.25%) in the gels had been chosen?

Authors’ response: Thank you for your question. We have added the following: “Preliminary studies of collagen-agarose hydrogels using 0.5% agarose with 0.2% or 0.5% collagen have shown that they possess pro-angiogenic properties (27, 28). Those with 0.2% agarose and 0.2% or 0.5% collagen were found to have poor angiogenesis (27, 28). However, these studies have shown limited angiogenesis characterization, with cell morphology being the predominant mode of evaluation. Moreover, stiffness of a 0.5% agarose and 0.5% collagen hydrogel blend is approximately 1.5 kPa (27, 28), while the stiffness of mammalian tracheal soft tissue in large animal and human cadaveric studies has been shown to range from 5-15 kPa (29, 30). In addition, higher hydrogel concentrations leads to a decrease in porosity (31), which can negatively impact processes of cellular migration and angiogenesis (32). Impeding these two processes within the studied hydrogel will ultimately lead to early tracheal graft failure and incomplete integration with host tissues  (32). In the design of our bioengineered composite tracheal graft, the mechanical support and structure is provided by the PCL support structure. The purpose of incorporating a hydrogel in this composite design is to eliminate the dead space between the ECM and PCL using a material that is maximally conducive to cellular migration and angiogenesis. Higher concentrations of collagen and agarose are likely to increase the viscosity of the hydrogel blend, limiting its ability to flow and completely occupy the volumetric deficits (including permeating through the perforated inter-ring PCL sheets) in our composite bioengineered tracheal graft. Thus, we developed an improved collagen-agarose hydrogel with lower concentrations of collagen and agarose than previously reported, with the aim to decrease porosity and increase cytocompatibility.”.

Comment 7: Line 11 Probably it would be better to specify which amount of collagen in hydrogel blend used in tracheal graft bioengineering is considered as low in the abstract as well.

Authors’ response: Thank you for your suggestion. We have added the following: “lower concentrations of collagen (< 5 mg/mL)”

Comment 8: Line 13 Probably it would be better to stress a bit more that the main goal is to create a collagen-agarose hydrogel blend which might be used in tracheal graft bioengineering and the obtained construct will withstand mechanical duress experienced by the native trachea. 

Authors’ response: Thank you. We have added the following to our Abstract: “The collagen-agarose hydrogel blend is meant to be cast around a three dimensional (3D) printed polycaprolactone support structure and wrapped in porcine small intestine submucosa ECM to create an off-the-shelf bioengineered tracheal implant.”.

Comment 9: Lines 16-17 In the Abstract it is described that ‘Fibroblasts were applied to the surface of the hydrogels, and endothelial cells were introduced within the hydrogels.’ And in the Materials and Methods section it is indicated that endothelial cells were mixed in a 3:1 ratio of fibroblasts to endothelial cells and mixed with the prepared hydrogel blends (lines 135-136). It would be better to clarify the procedure in the Abstract.

Authors’ response: Thank you for pointing out this discrepancy. The cells were indeed mixed for a homogenous cell solution, contrary to the Abstract.

The text in the Abstract was revised to accurately reflect the procedure used during cell culture.

“The hydrogel surface was seeded with fibroblasts, while both endothelial cells and fibroblasts (3:1 ratio) were mixed within the hydrogel matrix.”

Comment 10: Line 18 If the mixture of the cells was used, how were the endothelial cells determined, morphologically?

Authors’ response: Thank you for this suggestion. Due to restrictions on the word limit of the Abstract, have included this information in the Methods section.

Comment 11: Line 21 Would you please describe how the effectiveness of the gels was analyzed?

Authors’ response: Thank you for this suggestion. Due to restrictions on the word limit of the Abstract, have included this information in the Methods section.

Comment 12: Line 23 Would you please describe, how the elongated structures were assessed?

Authors’ response: Thank you for this suggestion. Due to restrictions on the word limit of the Abstract, have included this information in the Methods section.

Comment 13: Line 100 About the design of the composite tracheal graft – it is important to specify the current study goal and the final goal of the tracheal graft manufacture, it is a bit confusing. 

Authors’ response: Thank you for pointing out this potential point of confusion. We have modified the Introduction text to better separate how key design and target parameters (e.g., porosity and angiogenesis) of the studied hydrogel can affect the final goal of a tracheal graft (e.g., capable of integrating with host tissues).

“In addition, higher hydrogel concentrations leads to a decrease in porosity (31), which can negatively impact processes of cellular migration and angiogenesis (32). Impeding these two processes within the studied hydrogel will ultimately lead to early tracheal graft failure and incomplete integration with host tissues (32)

Comment 14: Lines 108-109 ‘we developed a novel collagen-agarose hydrogel with lower concentrations of collagen’ In the Abstract the following is indicated: ‘we created a novel hydrogel blend with higher concentrations of collagen’ (lines 12-13). It is confusing. Please clarify.

Authors’ response: Thank you for pointing out this point of confusion. The Abstract has been revised to be in-line with references included in the manuscript Discussion.

Specific text revisions to the Abstract include

“Collagen-agarose hydrogel blends currently used in tracheal graft bioengineering contain relatively high concentrations of collagen to withstand mechanical stresses associated with native trachea function (e.g., breathing). Unfortunately, the high collagen content restricts effective cell infiltration into the hydrogel.”

Comment 15: Lines 108-110 It is pointed out that Authors’ ‘developed a novel collagen-agarose hydrogel with lower concentrations of collagen and agarose than previously reported, with the aim to decrease porosity and increase cytocompatibility.’ If the data on porosity (microstructure) and cytocompatibility / cellular toxicity of the material will be provided (probably the references) it also would be a compliment.

Authors’ response: Thank you for bringing up this salient point pertaining to the ultimate purpose of replacing tracheal tissues.

Due to lack of funding and appropriate biomechanical testing equipment at our institute, we cannot perform biomechanical analyses (e.g., force-displacement testing, scanning electron microscopy of hydrogel pore sizes). As a surrogate metric, we can broadly comment that cell migration/elongation is correlated with hydrogel stiffness. For endothelial cells to migrate/elongate within a matrix on the scale of days, a hydrogel should roughly approximate 20 kPa to 1 MPa (Dessalles CA, Leclech C, Castagnino A, Barakat AI. Integration of substrate- and flow-derived stresses in endothelial cell mechanobiology. Commun Biol. 2021 Jun 21;4(1):764. doi: 10.1038/s42003-021-02285-w. PMID: 34155305; PMCID: PMC8217569.). The lower end of this range is comparable to human tracheal submucosa behavior (Teng Z, Trabelsi O, Ochoa I, He J, Gillard JH, Doblare M. Anisotropic material behaviours of soft tissues in human trachea: an experimental study. J Biomech. 2012 Jun 1;45(9):1717-23. doi: 10.1016/j.jbiomech.2012.04.002. Epub 2012 Apr 24. PMID: 22534565.).

Comment 16: Line 112 It would be better to decipher the following abbreviature: ‘SIS ECM’.

Authors’ response: Thank you for notifying us of this unexplained abbreviation.

We have replaced the abbreviation (not used anywhere else in the manuscript) with the full name of the product

“small intestine submucosa ECM”

Comment 17: Line 129 Would you please specify the gels of which concentrations were added to the wells.

Authors’ response: Thank you for suggesting a point of clarification. The Methods and Materials section has been revised to be more explicit

“200µL of each gel blend (1 mg/ml collagen + 0.25% agarose, 1 mg/ml collagen + 0.125% agarose, 2 mg/ml collagen + 0.25% agarose, 2 mg/ml collagen + 0.125% agarose) was added to the wells of a 48-well plate.”

Comment 18: Line 140 The analyses were performed most lately on day 7, why the cultures were maintained until day 14, would you please specify?

Authors’ response: Thank you for noticing our error. We only maintained our seeded hydrogels for 7 days, after which we preserved, processed, or froze them. We have fixed this in the Materials and Methods.

Comment 19: Line 141 Would you please explain how the assessment of cell morphology amongst the different hydrogel blends were performed, the parameters which had been analyzed, the program that had been used if any. 

Authors’ response: Thank you for indicating that the assessment of cell morphology could be explained upon further. Additional information was included in the Materials and Methods section

“Analysis was performed by a blinded observer who made qualitative observations of the images. Key parameters included comparative assessment of cell length (i.e., contiguous areas that are brighter/darker than the surrounding background) and elongation (i.e., deviation from spherical cell shape).”

Attempts to automate the process with ImageJ have been unsuccessful (not stated in the manuscript). Difficulties include changing hydrogel opacity (unable to discern background pixels across the whole image area) and inconsistent cell presentation (some cells brighter or darker than surrounding background pixels due to differences in elevation in 3D hydrogel).

Comment 20: Line 145 ‘5µL sections’ – Would you please specify, what do you mean?

Authors’ response: Thank you for noticing our error. We have corrected the units to micrometer (µm)

Comment 21: Line 145 ‘stained with hematoxylin and eosin (H&E).’ – Would the data be provided probably?

Authors’ response: Thank you for noticing our error. Hematoxylin and eosin staining was not performed on the sample slides. The Materials and Methods have been corrected to remove any mention of H&E staining.

Comment 22: Line 153 It would be better to indicate which samples were studied.

Authors’ response: Thank you for pointing out this potential source of confusion. We have clarified that the samples studied are simply high quality (i.e., defect free) sections.

“Select slides (i.e., sections with minimum defects) ...”

Comment 23: Lines 168-169 It would be better to describe a bit more the technique of how the distances between migrated cells and the nearest seeding surface were calculated with the software.

Authors’ response: Thank you for indicating this potential source of uncertainty. Additional text into the thresholding and segmentation protocol were included in the Materials and Methods section

“Cells were identified via intensity thresholding against the light hydrogel background. Afterward, the densely populated cell seeding surface was segmented from migrated cells by identifying large contiguous cell areas (i.e., greater than 15 µm2) indicating the contours of the collagen-agarose hydrogels”

Comment 24: Line 192 Figure 1 – It would be better if the microscopic images would be larger, probably would have larger magnification and resolution, and also indication of magnification or scale bars are required. Probably it would be better to remove the descriptions from the figure and add them into the figure legend or in the text. ‘Gel(G)’ is better to replace from the figure to the figure description. Would you please specify were the cells stained?

Authors’ response: Thank you for bringing up this salient point pertaining to the ultimate purpose of replacing tracheal tissues.

Due to lack of funding and appropriate biomechanical testing equipment at our institute, we cannot perform biomechanical analyses (e.g., force-displacement testing, scanning electron microscopy of hydrogel pore sizes). As a surrogate metric, we can broadly comment that cell migration/elongation is correlated with hydrogel stiffness. For endothelial cells to migrate/elongate within a matrix on the scale of days, a hydrogel should roughly approximate 20 kPa to 1 MPa (Dessalles CA, Leclech C, Castagnino A, Barakat AI. Integration of substrate- and flow-derived stresses in endothelial cell mechanobiology. Commun Biol. 2021 Jun 21;4(1):764. doi: 10.1038/s42003-021-02285-w. PMID: 34155305; PMCID: PMC8217569.). The lower end of this range is comparable to human tracheal submucosa behavior (Teng Z, Trabelsi O, Ochoa I, He J, Gillard JH, Doblare M. Anisotropic material behaviours of soft tissues in human trachea: an experimental study. J Biomech. 2012 Jun 1;45(9):1717-23. doi: 10.1016/j.jbiomech.2012.04.002. Epub 2012 Apr 24. PMID: 22534565.).

Comment 25: Line 200 Would you please specify how the ‘greater potential’ was determined?

Authors’ response: Thank you for pointing out this ambiguity in our Analysis. We have clarified the metric underlying the “greater potential”

“Amongst the constructs with 0.25% agarose, the hydrogel with 1mg/mL T1 collagen indicated greater potential for early vasculogenesis (i.e., minimal cell extension) than the hydrogel with 2mg/mL T1 collagen (i.e., no cell extensions observed) (Figure 3).”

Comment 27: What does that mean ‘minimal cell elongation’, ‘extensive cell elongations’, how these parameters had been determined?

Authors’ response: Thank you for indicating that the assessment of cell morphology could be explained upon further. “Cell elongation” was qualitatively assessed by a blinded observer asked to perform a comparative assessment of cell elongation (i.e., deviation from spherical cell shape).

Qualifying statements for the terms “minimal” and “extensive” have been added to the Results section for clarification

“More specifically, hydrogels composed of 0.125% agarose and 1-2 mg/ml T1 collagen produced the greatest (“extensive”) cell elongation based on blinded comparative analysis.”

“Of note, both 0.25% hydrogel formulations produced significantly reduced and inconsistent (“minimal”) cell elongation compared to the 0.125% hydrogel formulations as assessed during blinded comparative analysis.”

Comment 28: With which microscope the images were obtained?

Authors’ response: Thank you for pointing out this Materials and Method omission. The images were obtained using a Vectra automated imaging system (Akoya Biosciences). This information has been included in the manuscript.

“Cell morphology images were captured using a Vectra automated imaging system (Akoya Biosciences).”

“Slides were visualized using a Vectra automated imaging system (Akoya Biosciences)”

Comment 29: Line 203 Would you please specify which samples were analyzed?

Authors’ response: Thank you for your comment. At each mention, we have now stated which samples were analyzed:

“We selected one cross-section slide per gel blend for visualization and analysis, based on quality of tissue section and number of cells present.” 

“Select slides (i.e., sections with minimum defects) were also immunostained for alpha smooth muscle actin.”.

Comment 30: Line 227 ‘lower concentrations’ – It would be better to specify the ingredient.

Authors’ response: Thank you for indicating this point of potential ambiguity. We have indicated that lower concentrations of collagen and agarose were utilized in our study compared to the referenced papers.

We created an improved collagen-agarose hydrogel blend with lower concentrations of both biopolymers than previously studied (27, 28) and tested its angiogenic properties.

Comment 31: Line 230 Would you please specify, how the cellular attachment was studied?

Authors’ response: Thank you for providing us a chance to clarify our Discussion. Cellular attachment was studied via microscopy of Day 1 fibroblast migration. In Figure 1, hydrogels with 2 mg/ml T1 collagen showed lower cellular attachment (i.e., darkened pixels) when comparing 0.25% agarose to 0.125% agarose. This point is expanded upon in “Section 3.1. Fibroblast Migration”.

We hope this clarifies any lingering questions of how we arrived that this Discussion point.

Comment 32: Lines 230-233 It would be better to provide the data on the hydrogel blend structural integrity and degradation for 14 days period study, it would be interesting. Or, if possible, would you please give a reference to the research?

Authors’ response: Thank you for noticing our error. We only maintained our seeded hydrogels for 7 days, after which we preserved, processed, or froze them. We have fixed this in the Materials and Methods.

Comment 33: Line 238 Authors’ note that such factors as ‘composition, stiffness, and microstructure’ influence the angiogenic properties of the material, it would be a compliment if the information about the gels stiffness and microstructure would be characterized in the Article.

Authors’ response: Thank you for bringing up this salient point pertaining to the ultimate purpose of replacing tracheal tissues.

Due to lack of funding and appropriate biomechanical testing equipment at our institute, we cannot perform biomechanical analyses (e.g., force-displacement testing, scanning electron microscopy of hydrogel pore sizes). As a surrogate metric, we can broadly comment that cell migration/elongation is correlated with hydrogel stiffness. For endothelial cells to migrate/elongate within a matrix on the scale of days, a hydrogel should roughly approximate 20 kPa to 1 MPa (Dessalles CA, Leclech C, Castagnino A, Barakat AI. Integration of substrate- and flow-derived stresses in endothelial cell mechanobiology. Commun Biol. 2021 Jun 21;4(1):764. doi: 10.1038/s42003-021-02285-w. PMID: 34155305; PMCID: PMC8217569.). The lower end of this range is comparable to human tracheal submucosa behavior (Teng Z, Trabelsi O, Ochoa I, He J, Gillard JH, Doblare M. Anisotropic material behaviours of soft tissues in human trachea: an experimental study. J Biomech. 2012 Jun 1;45(9):1717-23. doi: 10.1016/j.jbiomech.2012.04.002. Epub 2012 Apr 24. PMID: 22534565.).

Comment 34: Line 320 It is important to specify in comparison with which concentration – with the studied in the current research or with the published in the papers – the lower concentration was considered?

Authors’ response: Thank you for your comment. We have added the following to our Aims sentence: “In this study, we characterize the capacity for fibroblast invasion and angiogenesis in this improved lower-concentration (compared to 5 mg/mL in aforementioned literature) gel blend.”

Comment 35: Lines 322-323 It would be better to present the information about the porosity of the gels in the text or probably a reference to the research.

Authors’ response: Thank you for bringing up this salient point pertaining to the ultimate purpose of replacing tracheal tissues.

Due to lack of funding and appropriate biomechanical testing equipment at our institute, we cannot perform biomechanical analyses (e.g., force-displacement testing, scanning electron microscopy of hydrogel pore sizes). As a surrogate metric, we can broadly comment that cell migration/elongation is correlated with hydrogel stiffness. For endothelial cells to migrate/elongate within a matrix on the scale of days, a hydrogel should roughly approximate 20 kPa to 1 MPa (Dessalles CA, Leclech C, Castagnino A, Barakat AI. Integration of substrate- and flow-derived stresses in endothelial cell mechanobiology. Commun Biol. 2021 Jun 21;4(1):764. doi: 10.1038/s42003-021-02285-w. PMID: 34155305; PMCID: PMC8217569.). The lower end of this range is comparable to human tracheal submucosa behavior (Teng Z, Trabelsi O, Ochoa I, He J, Gillard JH, Doblare M. Anisotropic material behaviours of soft tissues in human trachea: an experimental study. J Biomech. 2012 Jun 1;45(9):1717-23. doi: 10.1016/j.jbiomech.2012.04.002. Epub 2012 Apr 24. PMID: 22534565.).

Comment 36: Please check, the references should be described according to the recommendations published on the Journal’s website.

Authors’ response: Thank you. We have ensured that the References are in the journal-stipulated format.

Reviewer 3

Dear Authors’, 

Comment 1: The manuscript entitled "Novel Composite Hydrogel for Bioengineered Tracheal Graft demonstrates Effective Early Angiogenesis" represents a very optimistic idea in the field and the contained informaton will be very valuable for the readers of the issue. However, major revisions are required before the manuscript is further processed.

Authors’ response: Thank you for your optimistic viewpoint of our work and for the critical feedback to elevate our manuscript to the journal publication standards. We hope that the following revisions will address your concerns.

Comment 2: Do the Authors’ think that the hydrogel bioengineered scaffolds would be characterized by the similar biomechanical properties as the native or even decellularized tissue. Is it efficient to perform biomechanical analysis of the hydrogels.

Authors’ response: Thank you for bringing up this salient point pertaining biome to the ultimate purpose of replacing tracheal tissues.

Due to lack of funding and appropriate biomechanical testing equipment at our institute, we cannot perform biomechanical analyses (e.g., force-displacement testing, scanning electron microscopy of hydrogel pore sizes). As a surrogate metric, we can broadly comment that cell migration/elongation is correlated with hydrogel stiffness. For endothelial cells to migrate/elongate within a matrix on the scale of days, a hydrogel should roughly approximate 20 kPa to 1 MPa (Dessalles CA, Leclech C, Castagnino A, Barakat AI. Integration of substrate- and flow-derived stresses in endothelial cell mechanobiology. Commun Biol. 2021 Jun 21;4(1):764. doi: 10.1038/s42003-021-02285-w. PMID: 34155305; PMCID: PMC8217569.). The lower end of this range is comparable to human tracheal submucosa behavior (Teng Z, Trabelsi O, Ochoa I, He J, Gillard JH, Doblare M. Anisotropic material behaviours of soft tissues in human trachea: an experimental study. J Biomech. 2012 Jun 1;45(9):1717-23. doi: 10.1016/j.jbiomech.2012.04.002. Epub 2012 Apr 24. PMID: 22534565.).

Comment 3: For the angiogenesis assay, I would expect the use of endothelial cells instead of fibroblasts.

Authors’ response: Thank you for the constructive comment. While endothelial cells are the major cells involved in angiogenesis (e.g., formation of blood vessels), fibroblasts play a supporting role in the process by locally producing key factors (e.g., fibroblast growth factor) which assist with the process. Hence, we examined fibroblast migration as a complementary function in angiogenesis since fibroblast infiltration should match endothelial cell growth for optimal hydrogel vascularization. This interplay between endothelial cell and fibroblast function is expanded upon in the Introduction

“These fibroblasts support neovascularization and synthesize new ECM by producing matrix proteins, growth factors, proteases, and other biochemical modulators. Fibroblasts further enhance neovascularization by initiating vasodilatory expansion and improving structural integrity of the new vasculature (15). Co-cultures of endothelial cells and fibroblasts are commonly used in in-vitro angiogenesis assays and provide an established method of testing pro-angiogenic capabilities (16)”

Comment 4: In Figure 3, please provide images with higher magnification and better resolution, to demonstrate better the cell distribution among the different hydrogel grafts.

Authors’ response: Thank you. We have improved the figure quality and added scale bars.

Reviewer 4

Comment 1: Martins et al showed an improved composite of hydrogel for tracheal graft research in this manuscript. The topic is of interest in translational medicine, and the study design is largely appropriate, but the three are major concerns regarding part of the results that must be addressed.

Authors’ response: Thank you for your optimistic viewpoint of our work and for the critical feedback to elevate our manuscript to the journal publication standards. We hope that the following revisions will address your concerns.

Comment 2: It is critical to demonstrate the cells are alive in the hydrogel and provide counts of the numbers of cells. The images in Figure 1, as well as the IF images in figure S2 are of very low quality. Higher quality images and a clear description of the methods must be provided.

Authors’ response: Thank you for the input regarding the veracity of our cytocompatibility claims and quality of our images.

We encountered difficulty when attempting to directly count the number of viable cells using traditional methods (e.g., MTT assay, LIVE/DEAD staining) due to several limitations of the hydrogel culture system (e.g., poor diffusion of viability testing compound throughout the hydrogel matrix, collagen and agarose autofluorescence, hydrogel thickness, and inability to focus imaging plane on all cells in 3D space). Instead, we rely on cell migration and elongation behaviors over the 7 days of culture as indirect metrics of cell viability within the hydrogel matrix. Greater discussion of the correlation between cell migration/elongation and cell viability have been included in the Results section.

“In addition, the demonstration of fibroblast migration within the hydrogel matrix indicates that the collagen-agarose blend: (1) is biocompatible, (2) displays cell attachment motifs to allow cell motility, and (3) remains sufficiently porous to permit nutrient diffusion and cell infiltration.”

“In addition, the demonstration of early vasculogenesis in select collagen-agarose concentrations further supports key hydrogel biological and mechanical characteristics including biocompatibility (cells remain viable enough to undergo vasculogenesis), cell attachment motifs (permit cells to form focal adhesions during the elongation and vasculogenesis process), and porosity (ensure sufficient nutrient/waste exchange occurs during the 7-day culture period).”

Higher quality microscopy images have been included in the figures.

Comment 3: The gene expression changes shown in Table 3 are not scientifically sound. It does not seem possible that gene ACTA2 expression would change 40+ folds just by increasing collagen concentration from 1mg/mL to 2 mg/mL. Likewise, endothelial markers CDH5, TIE1 also changed 4+ folds just by manipulating collagen or agarose concentration. It looks that these dramatic changes are more likely caused by drastically different conditions of cells in each treatment group. Did the Authors’ use an internal control to normalize the PCR?

Authors’ response: Thank you for noticing this significant omission. We used B2M (β-2-microglobulin) as a housekeeping gene. This has been mentioned in our Methodology: “Beta-2 microglobulin was used as the housekeeping gene and served to standardize the results of the RNA analysis.”.

Regarding the magnitude of gene expression change, these values are within the ranges reported in other publications. For example, Zhang et al. (Harnessing 3D collagen hydrogel-directed conversion of human GMSCs into SCP-like cells to generate functionalized nerve conduits, npj Regenerative Medicine, 2021) reported significant relative mRNA expression changes for Gdnf from ~20 at 2 mg/ml collagen to ~45 at 3mg/ml collagen. While there are significant differences between culture conditions, tested collagen concentrations, and target genes, our report (Table 3, 0.125% agarose, ACTA2: +18.64 at 1 mg/ml collagen vs +47.50 at 2 mg/ml collagen) broadly conforms to the magnitude of gene expression changes seen in other publications.

Comment 4: Due to the concerns stated above, the findings in this manuscript cannot support the claim that "Hydrogel blends with collagen and low 25 agarose concentrations are effective in allowing early cellular infiltration and angiogenesis, making such gels a suitable cell substrate for use in the development of composite bioengineered tracheal grafts."

Authors’ response: Thank you for your comment. We have reworded our statement to make it less strong: “Hydrogel blends with collagen and low agarose concentrations may be effective in allowing early cellular infiltration and angiogenesis, making such gels a suitable cell substrate for use in the development of composite bioengineered tracheal grafts.”.

Comment 5: The composition of collagen and agarose is not novel just by altering the concentrations. It is appropriate to change "Novel" to "Improved" in the title as well as in the text.

Authors’ response: Thank you for the note on the extent of our manuscript’s claims. We have modified instances of “novel” hydrogel blends/composition to read “improved”. For example, the title now reads “Improved Composite Hydrogel for Bioengineered Tracheal Graft demonstrates Effective Early Angiogenesis”

Comment 6: Although the title includes “tracheal graft”, but there is no experiment on the implantation in vivo.  It is better to remove the “tracheal graft”. The Authors’ can discuss the application potential in the discussion section.

Authors’ response: Thank you for your suggestion. We have amended the title to: “Improved Composite Hydrogel demonstrates Effective Early Angiogenesis: Implications for Development of a Bioengineered Tracheal Graft”.

Reviewer 2 Report

Comments and Suggestions for Authors

The Article is devoted to the study of novel collagen-agarose-based hydrogel blends which will help in the development of bioengineered tracheal grafts. The study is interesting, important, and great efforts were put into its realization.

Authors studied the newly created materials for fibroblast invasion and angiogenesis for 7 days. Interesting that the co-cultures of human umbilical vein endothelial cells and dermal fibroblasts were used during the research.

The limitations of the study are characterized which is a benefit.

The following comments do not diminish the value of the Article:

Line 3 Probably it would be better to specify in the Article the criteria that might be used to assess the effectiveness of angiogenesis? As this term is used in the title of the Article.

Would you please specify why the current concentrations of collagen (1 mg/mL and 2 mg/mL) and agarose (0.125% and 0.25%) in the gels had been chosen?

Line 11 Probably it would be better to specify which amount of collagen in hydrogel blend used in tracheal graft bioengineering is considered as low in the abstract as well.

Line 13 Probably it would be better to stress a bit more that the main goal is to create a collagen-agarose hydrogel blend which might be used in tracheal graft bioengineering and the obtained construct will withstand mechanical duress experienced by the native trachea. 

Lines 16-17 In the Abstract it is described that ‘Fibroblasts were applied to the surface of the hydrogels, and endothelial cells were introduced within the hydrogels.’ And in the Materials and Methods section it is indicated that endothelial cells were mixed in a 3:1 ratio of fibroblasts to endothelial cells and mixed with the prepared hydrogel blends (lines 135-136). It would be better to clarify the procedure in the Abstract.

 Line 18 If the mixture of the cells was used, how were the endothelial cells determined, morphologically?

Line 21 Would you please describe, how the effectiveness of the gels was analyzed?

Line 23 Would you please describe, how the elongated structures were assessed?

Line 100 About the design of the composite tracheal graft – it is important to specify the current study goal and the final goal of the tracheal graft manufacture, it is a bit confusing. 

Lines 108-109 ‘we developed a novel collagen-agarose hydrogel with lower concentrations of collagen’ In the Abstract the following is indicated: ‘we created a novel hydrogel blend with higher concentrations of collagen’ (lines 12-13). It is confusing. Please clarify.

Lines 108-110 It is pointed out that Authors ‘developed a novel collagen-agarose hydrogel with lower concentrations of collagen and agarose than previously reported, with the aim to decrease porosity and increase cytocompatibility.’ If the data on porosity (microstructure) and cytocompatibility / cellular toxicity of the material will be provided (probably the references) it also would be a compliment.

Line 112 It would be better to decipher the following abbreviature: ‘SIS ECM’.

Line 129 Would you please specify the gels of which concentrations were added to the wells.

Line 140 The analyses were performed most lately on day 7, why the cultures were maintained until day 14, would you please specify?

Line 141 Would you please explain how the assessment of cell morphology amongst the different hydrogel blends were performed, the parameters which had been analysed, the program that had been used if any. 

Lin 145 ‘5µL sections’ – Would you please specify, what do you mean?

Line 145 ‘stained with hematoxylin and eosin (H&E).’ – Would the data be provided probably?

Line 153 It would be better to indicate which samples were studied.

Lines 168-169 It would be better to describe a bit more the technique of how the distances between migrated cells and the nearest seeding surface were calculated with the software.

Line 192 Figure 1 – It would be better if the microscopic images would be larger, probably would have larger magnification and resolution, and also indication of magnification or scale bars are required. Probably it would be better to remove the descriptions from the figure and add them into the figure legend or in the text. ‘Gel(G)’ is better to replace from the figure to the figure description. Would you please specify were the cells stained?

Line 200 Would you please specify how the ‘greater potential’ was determined?

Line 202 Figure 3 – It would be better if the microscopic images would be a bit larger, probably would have larger magnification and resolution, and also indication of magnification or scale bars are required. Probably it would be better to remove the descriptions from the figure and add them into the figure legend or in the text.

What does that mean ‘minimal cell elongation’, ‘extensive cell elongations’, how these parameters had been determined?

With which microscope the images were obtained?

Line 203 Would you please specify which samples were analysed?

Lines 204-206 Probably it would be better to specify the sentence according  to the following corrections.

Line 227 ‘lower concentrations’ – It would be better to specify the ingredient.

Line 230 Would you please specify, how the cellular attachment was studied?

Lines 230-233 It would be better to provide the data on the hydrogel blend structural integrity and degradation during 14 days period study, it would be interesting. Or, if possible, would you please give a reference to the research?

Line 238 Authors note that such factors as ‘composition, stiffness, and microstructure’ influence the angiogenic properties of the material, it would be a compliment if the information about the gels stiffness and microstructure would be characterized in the Article.

Line 320 It is important to specify in comparison with which concentration – with the studied in the current research or with the published in the papers – the lower concentration was considered?

Lines 322-323 It would be better to present the information about the porosity of the gels in the text or probably a reference to the research.

Please check, the references should be described according to the recommendations published on the Journal’s website.

Author Response

(The authors gave the same response as above.)

Reviewer 3 Report

Comments and Suggestions for Authors

Dear Authors, 

The manuscript entitled "Novel Composite Hydrogel for Bioengineered Tracheal Graft demonstrates Effective Early Angiogenesis" represents a very optimistic idea in the field and the contained informaton will be very valuable for the readers of the issue. However, major revisions are required before the manuscript further processed.

1) Do the authors think that the hydrogel bioengineered scaffolds would be characterized by the similar biomechanical properties as the native or even decellularized tissue.

2) Is it efficient to perform biomechanical analysis of the hydrogels.

3) For the angiogenesis assay, i would expect the use of endothelial cells instead of fibroblasts.

4) In Figure 3, please provide images with higher magnification and better resolution, to demonstrate better the cell distribution among the different hydrogel grafts.

Author Response

(The authors gave the same response as above.)

Reviewer 4 Report

Comments and Suggestions for Authors

Matins et al showed an improved composite of hydrogel for tracheal graft research in this manuscript. The topic is of interest in translational medicine, and the study design is largely appropriate, but the there are major concerns regarding part of the results that must be addressed.

Major concerns:

1. It is critical to demonstrate the cells are alive in the hydrogel, and provide counts of the numbers of cells. The images in Figure 1, as well as the IF images in figure S2 are of very low quality. Higher quality images and a clear description of the methods must be provided.

2. The gene expression changes showed in Table 3 are not scientifically sound. It does not seem possible that gene ACTA2  expression would change 40+ folds just by increasing collagen concentration from 1mg/mL to 2 mg/mL. Likewise, endothelial markers CDH5, TIE1 also changed 4+ folds just by manipulating collagen or agarose concentration. It looks that these dramatic change are more likely caused by drastically different conditions of cells in each treatment group. Did the authors use an internal control to normalize the PCR? 

3. Due to the concerns stated above, the findings in this manuscript  can not support the claim that "Hydrogel blends with collagen and low 25 agarose concentrations are effective in allowing early cellular infiltration and angiogenesis, mak- 26 ing such gels a suitable cell substrate for use in the development of composite bioengineered tra- 27 cheal grafts."

Minor concerns:

1, The composition of collagen and agarose is not novel just by altering the concentrations. It is appropriate to change "Novel" to "Improved" in then title as well as in the text.

2, Although the title includes “tracheal graft”, but there is no experiment on the implantation in vivo.  It is better to remove the “tracheal graft”. The authors can discuss the application potential in the discussion section. 

Author Response

(The authors gave the same response as above.)

Round 2

Reviewer 3 Report

Comments and Suggestions for Authors

Dear Authors,

The majority of my comments have been well addressed by the authors.

Well done!!

Author Response

Comment: Dear Authors, The majority of my comments have been well addressed by the authors. Well done!!

Authors' Response: Thank you for your review and feedback!

Reviewer 4 Report

Comments and Suggestions for Authors

The authors addressed most of my concerns, except my comment #3 regarding the gene expression changes.

I suggest the authors to repeat the qPCR experiments to validate this claim.

Author Response

Comment: The authors addressed most of my concerns, except my comment #3 regarding the gene expression changes. I suggest the authors to repeat the qPCR experiments to validate this claim.

Authors' Response: Thank you for your review. Please note that we have already performed technical and biological replication to arrive at the results presented in our manuscript. We have attached the raw data from the final level of analysis for the ACTA2 expression levels for your information. As stated in our previous response document, the levels of ACTA2 expression fall within the ranges of that reported in the literature. We have mentioned all the above in our Manuscript main text as well.
